# Estimating an affine term structure model of interest rates with correlated noise

Shu WU[1], Rende Li[2]*

**1** Sino-German College, University of Shanghai for Science and Technology, Shanghai, PR China, **2** Library, University of Shanghai for Science and Technology, Shanghai, PR China

* lirende@usst.edu.cn

## Abstract

Kalman filtering for the affine term structure model of interest rates is typically applied under the assumption of white noise. However, correlated noise frequently occurs during actual data processing. The accuracy and reliability of the filter are compromised if the correlated noise is assumed to be white noise. This paper develops a measurement expansion scheme for the affine term structure model based on the whitening properties of the Kalman filter, enabling latent factor estimation under the general assumption of correlated noise. The simulation results indicate that the estimation based on the measurement expansion scheme achieves higher accuracy compared to the traditional method.

**Data Availability Statement:** The simulation data supporting the findings of this study are available in GitHub at https://github.com/lirende29/Affine-Term-Structure/.

**Funding:** The author(s) received no specific funding for this work.

## 1. Introduction

In China, the bond market is relatively small, with limited bond types available. Bond trading on exchanges began in 1990. Bonds began trading on the Shanghai Stock Exchange in 1990, followed shortly by the launch of the Shenzhen Stock Exchange in 1991 [1]. Deposit interest rates are commonly used as benchmarks when referring to market rates. However, bond rates are determined by the market. Bond prices respond more quickly to economic changes. With the ongoing reform of interest rate liberalization in China, the term structure in the bond market has become increasingly representative of market interest rates.

The term structure of interest rates is the dependence between the interest rates and the time to maturity [2]. A large body of literature on dynamic term structure modeling is distinguished by the specification of the underlying processes used [3]. One of the most popular classes of term structure models is the affine class. The primary advantage of this class is its analytical tractability, including the availability of closed-form solutions for zero-coupon bond prices [4]. Recent studies have further developed these models. For instance, Christensen and Rudebusch (2020) explored affine term structure models in the context of a zero lower bound world, enhancing our understanding of interest rate behaviors in constrained environments [5]. Adrian, Crump, and Moench (2019) provided insights into pricing the term structure with linear regressions, offering new methodologies for empirical analysis [6]. Moreover, Joslin, Singleton, and Zhu (2021) introduced a new perspective on Gaussian dynamic term structure models, contributing to the theoretical advancements in this field [7].

**Competing interests:** NO authors have competing interests.

The single-factor linear model proposed by Vasicek is a Gaussian affine model, where all state variables have constant volatilities [8]. In contrast, CIR affine models assume CIR-type square root volatilities for all state variables [9]. In both Vasicek-type and CIR-type affine models, two assumptions are imposed: the state variables are independent and the price of risk is a multiple of interest rate volatility [10]. However, estimation results reveal limitations within these models. Dai and Singleton find that relaxing the correlation restrictions leads to a model that passes several goodness-of-fit tests over their sample period [11]. Duffee develops the "essentially affine" models which not only retain the tractability of standard models, but also allow compensation for interest rate risk to vary independently of interest rate volatilities. This additional flexibility proves useful in forecasting future yields [12]. And also in this paper, the empirical results document that the pure Gaussian model of the essentially affine ones can generate reasonable forecasts of future yields in the sense that the predictive power of the term structure is subsumed within the model's forecasts [12].

Within the affine term structure setup, usually two submodels are included. One is to determine the time series movements of the yield curve driven by the latent factors (time-series dimension) and the other is to reflect the shape of the curve at a given time using a function of the latent factors and the time to maturity (cross-sectional dimension). In the numerical analysis, we use the "panel data" approach, where the data of both dimensions are taken into consideration to fully exploit the restrictions imposed by the affine model itself. Only by combining data from both dimensions can we infer the crucial parameter of the market price of risk, a key factor in pricing interest rate derivatives. A natural way to deal with panel data with the latent feature is Kalman filter which is also commonly believe to be the most efficient method of the estimation of unobserved variables [13–17]. Not surprisingly, many recent studies have been conducted along this line of thought [18–21].

Usually, there are more observed zero coupon yields than latent factors, so the term structure model itself cannot explain all the variations in the data. Thus, some form of measurement noise, which originate from possible non-synchronous trading, rounding of prices, and bid-ask spreads, is necessary. In most literature using Kalman fitler to get the interest rate term structure's estimation, the structure of measurement noise occurring in the state space form required by Kalman fitler approach is often assumed to be independent and identically distributed (i.i.d) for computational convenience [22–28]. However, Geyer and Pichler find that the results of diagnostic checking provide strong evidence against the adequacy of multifactor CIR models for the residuals are strongly biased and autocorrelated [29]. When analyzing the term structure using affine models, De Jong finds that both strong time serial and cross-sectional correlation exist in the measurement noise [14]. More recently, Dempster and Tang examine affine models for both bond yields and commodity futures and find that measurement noise mainly comes from serial correlated influence [19]. Thus i.i.d assumption is violated. The purpose of this paper is to propose an estimation solution based on Kalman filter method to alleviate this problem.

Recent advancements in the field of state estimation and Kalman filtering have introduced sophisticated techniques to address challenges associated with correlated and non-Gaussian noise, as well as applications in complex financial systems. For instance, adaptive Kalman filters have evolved to accommodate correlated noise structures and dynamically adjust to changes in noise characteristics, significantly enhancing accuracy in real-world applications such as tracking and sensor networks. Advances in Bayesian filtering have also expanded the applicability of state estimation models to non-Gaussian noise conditions, as seen in the development of Cubature Kalman filters and particle filtering techniques, which enable robust estimation even under heavy-tailed or asymmetric noise distributions. In financial contexts, recent literature has explored nonlinear and non-Gaussian state-space modeling to better

capture the complexities of financial data, such as yield curve dynamics and market volatility. These models incorporate structural breaks and regime changes, reflecting more realistic economic conditions. Moreover, enhancements to classic works on Kalman filtering have integrated adaptive frameworks that allow for real-time adjustments in response to dynamic noise profiles, thereby improving robustness and relevance in applications such as financial forecasting and risk management. These innovations mark significant progress in both theoretical and practical applications of Kalman filtering, offering greater flexibility and resilience in dynamic, noisy environments.

The remainder of the paper is composed as follows. Section 2 describes the theoretical model and its implementation. Section 3 offers a novel state space form of the term structure model of interest rates and the corresponding illustration of the Kalman filter method based on the measurement expand scheme. Section 4 discusses the numerical results. The main conclusions are stated at the end.

## 2. Discretization of the essentially Gaussian affine model

In the affine model, the short rate $r_t$ at time $t$ is described by [30]

$$r_t = \delta_0 + I^T x_t$$

In this paper, the state variables are denoted as $x_t = (x_{1t} x_{2t})^T$. $\delta_0$ is a scalar and $I^T$ is the unit vector. Under the equivalent martingale measure, for $i = 1, 2$, the dynamics of the state variables are governed by the stochastic differential equation [31]

$$dx_{it} = -k_i x_{it} dt + \sigma_i dz_{it}^\phi$$

with $dz_{it}^\phi$ being independent Wiener process. The parameter $k_i$ controls the degree of mean reversion. The parameter $\sigma_i$ is the volatility, or diffusion coefficient, of $x_{it}$.

The bond's factor model is completed by specifying the dynamics of $x_t$, or rather, of the market price of risk, under the physical measure. Denote the state price deflator at time $t$ by $\pi_t$, we get

$$\frac{d\pi_t}{\pi_t} = -r_t dt - \lambda_t dz_t$$

Element $i$ of the two-vector $\lambda_t$ represents the market price of risk associated with $dz_{it}$ which is the corresponding Wiener process and satisfies $dz_{it} = \epsilon_{it}\sqrt{dt}$, $\epsilon_{it} \sim N(0, 1)$, represented without the tildes, under the physical measure. In this paper, we use the following form for the market price of risk [32]

$$\lambda_t = \lambda_1 + \lambda_2 x_{it}$$

Here, $\lambda_1$ represents the time-invariant part in the risk premium and $\lambda_2$ the time-variant part. This enables us to write the dynamics of $x_t$ under the physical measure as follows

$$dx_{it} = -k_i x_{it} dt + \sigma_i dz_{it}^\phi = -k_i x_{it} dt + \sigma_i(\lambda_{1t} dt + dz_{it}) = \begin{bmatrix} (k_i - \sigma_i\lambda_{12}) & (\sigma_i\lambda_{11}) \\ & -x_t \\ k_i - \sigma_i\lambda_{22}) & \end{bmatrix} dt + \sigma_i dz_{it} \ (5)$$

$$\text{For } i = 1, 2, \text{ let } K = \begin{bmatrix} k_1 - \sigma_1 \lambda_{12} & 0 \\ 0 & k_2 - \sigma_2 \lambda_{22} \end{bmatrix}, \Theta = \begin{bmatrix} \Theta_1 \\ \Theta_2 \end{bmatrix} = \begin{bmatrix} \sigma_1 \lambda_{11} & k_1 - \sigma_1 \lambda_{12} \\ \sigma_2 \lambda_{21} & k_2 - \sigma_2 \lambda_{22} \end{bmatrix},$$

$$x_t = \begin{bmatrix} x_{1t} \\ x_{2t} \end{bmatrix}, \sigma = \begin{bmatrix} \sigma_1 & 0 \\ 0 & \sigma_2 \end{bmatrix}, \sigma_1 = \sigma_1, \sigma_2 = \sigma_2, dz_t = \begin{bmatrix} dz_{1t} \\ dz_{2t} \end{bmatrix}, \text{ and we get}$$

$$dx_t = K(\Theta - x_t)dt + \sigma dz_t \tag{6}$$

In the essentially Gaussian affine model, to estimate the unobservable state variables from the yields observed at discrete time intervals, we use the state space system, which includes a transition equation and a measurement equation, needed for Kalman filter estimation. Since we are estimating a continuous time model using discretely sampled observations, the transition Eq (6) should be first rewritten into a discrete form [33]. By applying an Euler discretization to (6), we get

$$x_{t+h} = (1 - Kh)x_t + K\Theta h + \sigma \sqrt{h} \epsilon_t \tag{7}$$

Here, $\{\epsilon_t\}$ is white noise series, $\epsilon_t \sim NID(0, 1)$ and $h$ denotes the sampling interval. $E(\epsilon_t) = 0$ and

$$\text{Cov}[\epsilon_{s,t}] = E[\epsilon_s \epsilon_t^T] = Q_\epsilon \delta_{st} \text{ where } \delta_{st} \text{ is a Kronecker Delta function with } \delta_{st} = \begin{cases} 1, & \text{if } s = t \\ 0, & \text{if } s \neq t \end{cases}.$$

The price $P(x_t, \tau)$ and yield $y(x_t, \tau)$ of a zero coupon bond that matures at time $T$ are

$$P(x_t, \tau) = \exp[A(\tau) + B(\tau)^T x_t] \tag{8}$$

Here, $\tau = T - t$, $A(\tau)$ is a scalar function, $B(\tau) = (B_1(\tau)B_2(\tau))$ and the yield $y(x_t, \tau)$ considered as a function of $\tau$ will be referred to as the term structure at time $t$. The measurement Eq (9) can be expressed in the state space form by adding noise term to the equations for the observable bond rates:

$$y(x_{t+h}, \tau) + \frac{A(\tau)}{\tau} = -\frac{B(\tau)^T}{\tau} x_{t+h} + \eta_{t+h} \tag{10}$$

where $\eta_t = (\eta_{1t}, \ldots, \eta_{Mt})^T$ and $M$ means the number of observations at time $t$. Each noise term is introduced to allow for small noise and imperfections in the observed bond rates. Following Chen and Scott, we allow serial correlation and contemporaneous correlation between the measurement noise [13]. The serial correlation is modeled as a first-order autoregressive process [13]

$$\eta_{j,t+h} = \rho_j \eta_{jt} + e_{j,t+h} \tag{11}$$

where $j = 1, \ldots, M$, $\{e_t\}$ is white noise series with $E(e_t) = 0$ and $E[e_t e_t^T] = \Omega \delta_{st}$, and $\rho_j$ is the $j$th diagonal element of the diagonal matrix $\rho$, with $-1 \leq \rho \leq 1$.

The model described up to this point is an exact discrete time representation of the two-factor essentially Gaussian affine model. Both (7) and (10) form the discrete time state space system and (11) indicates a structure of the measurement noise.

As to (8), the functions $A$ and $B$ can be expressed in a closed form and the problem solving process will be explained in the remaining part of this section.

With Ito lemma applied, we can get from (6) and (8) that [34]

$$dP = \left[ \sum_{i=1}^{2} \left( \frac{\partial P}{\partial x_{it}} K_i(\Theta_i - x_{it}) \right) + \frac{\partial P}{\partial t} + \frac{1}{2} \sum_{i=1}^{2} \left( \frac{\partial^2 P}{\partial x_{it}^2} \sigma_i^2 \right) \right] dt + \sum_{i=1}^{2} \left( \frac{\partial P}{\partial x_{it}} \sigma_i dz_{it} \right) \tag{12}$$

The price of any bond measured in units of the value of the state price deflator follows a martingale, that is [35]

$$P_t = E_t \left( \frac{\pi_T}{\pi_t} \right) \tag{13}$$

Multiply both sides of (13) with $\pi_t$, we get

$$P_t \pi_t = E_t(\pi_T P_T) \tag{14}$$

From one of the martingale properties, we know [36]

$$E_t[d(P_t \pi_t)] = 0 \tag{15}$$

Substitute $dP$ with (12), $d\pi$ with (3) and ignore the high order infinitesimal terms, then

$$
\begin{aligned}
d[P_t \pi_t] &= P_t d\pi_t + \pi_t dP_t + dP_t d\pi_t \\
&= \pi_t \left[ \frac{\partial P}{\partial t} + \sum_{i=1}^{2} \left( \frac{\partial P}{\partial x_{it}} K_i(\Theta_i - x_{it}) + \frac{1}{2} \frac{\partial^2 P}{\partial x_{it}^2} \sigma_i^2 \right) \right] dt \\
&\quad - P_t r_t \pi_t dt - \lambda_t \pi_t \sum_{i=1}^{2} \frac{\partial P}{\partial x_{it}} \sigma_i dz_{it} + o(dt)
\end{aligned}
\tag{16}
$$

This equation can be simplified with the help of (15). This yields

$$\pi_t \left[ \frac{\partial P}{\partial t} + \sum_{i=1}^{2} \left( \frac{\partial P}{\partial x_{it}} K_i(\Theta_i - x_{it}) + \frac{1}{2} \frac{\partial^2 P}{\partial x_{it}^2} \sigma_i^2 \right) \right] - P_t r_t \pi_t - \lambda_t \pi_t \sum_{i=1}^{2} \frac{\partial P}{\partial x_{it}} \sigma_i = 0 \tag{17}$$

and consequently

$$P_t r_t = \sum_{i=1}^{2} \left[ \frac{\partial P}{\partial x_{it}} K_i(\Theta_i - x_{it}) + \frac{1}{2} \frac{\partial^2 P}{\partial x_{it}^2} \sigma_i^2 \right] + \frac{\partial P}{\partial t} - \lambda_t \sum_{i=1}^{2} \frac{\partial P}{\partial x_{it}} \sigma_i \tag{18}$$

The partial derivatives of $P_t$ with respect to $x_{it}$ and $t$ can be obtained from (8). At the same time, substitute $r_t$ with (1) and then the functions of $A(\tau)$ and $B(\tau)$ can be calculated numerically by solving a series of ordinary differential equations (ODEs) with initial conditions $A(0) = 0$ and $B(0) = 0$, where

$$
\begin{aligned}
A(\tau) = \sum_{i=1}^{2} &\left[ \frac{K_i \Theta_i - \sigma_i \lambda_{1i}}{K_i + \sigma_i \lambda_{2i}} \left( 1 - e^{-(K_i + \sigma_i \lambda_{2i})\tau} \right) - \tau \right. \\
&\left. + \frac{\sigma_i^2}{2(K_i + \sigma_i \lambda_{2i})^2} \left( \frac{-3 e^{-(K_i + \sigma_i \lambda_{2i})\tau}}{2(K_i + \sigma_i \lambda_{2i})} + \tau + \frac{2 e^{-(K_i + \sigma_i \lambda_{2i})\tau}}{K_i + \sigma_i \lambda_{2i}} \right) \right]
\end{aligned}
\tag{19}
$$

$$B_i(\tau) = \frac{e^{-(K_i + \sigma_i \lambda_{2i})\tau} - 1}{K_i + \sigma_i \lambda_{2i}}, \quad i = 1, 2 \tag{20}$$

## 3. The state space setup

For the state space system, we assume that the statistical properties of $\{\epsilon_t\}$, $\{\eta_t\}$, the initial state $x_0$, the initial noise $\eta_0$ and the covariance matrix $V$ of the prediction error are as follows: $E[\epsilon_t \epsilon_t^T] = 0$, $E[x_0] = \mu_0$, $V[x_0] = \bar{P}_0$, $E[\eta_t] = 0$, $V[\eta_0] = R_0$, $E[x_0 \eta_0^T] = 0$, $E[x_0 \epsilon_t^T] = 0$, $E[\eta_0 \epsilon_t^T] = 0$.

If the correlated noise appears in the transition equation, the common practice is to adopt the method of state vector expanding. When the noise is correlated in the measurement equation, if we still use the state vector expanding method, then, for example, in our case, the state space system after the expanding method is

$$\begin{bmatrix} x_{t+h} \\ \eta_{j,t+h} \end{bmatrix} = \begin{bmatrix} 1 - Kh & 0 \\ 0 & \rho \end{bmatrix} \begin{bmatrix} x_t \\ \eta_t \end{bmatrix} + \begin{bmatrix} K\Theta h \\ 0 \end{bmatrix} + \begin{bmatrix} \sigma\sqrt{h} & 0 \\ 0 & 1 \end{bmatrix} \begin{bmatrix} \epsilon_t \\ e_{j,t+h} \end{bmatrix} \tag{21}$$

$$y(x_{t+h}, \tau) + \frac{A(\tau)}{\tau} = -\frac{B(\tau)^T}{\tau} \begin{bmatrix} x_{t+h} \\ \eta_{t+h} \end{bmatrix} \tag{22}$$

It seems that we've got a new state space system which satisfies the assumptions for Kalman filter scheme. However, the new conflict emerges because the measurement error covariance matrix for (22) is now a zero matrix, which is not permitted in the algorithm of Kalman filter. To ensure the existence of the Kalman gain matrix, the measurement error covariance matrix must be positive definite. We propose in this section a new state space form for the term structure model based on the measurement expand scheme to address the problem. The main point of this method is to make some change on the measurement equation to get the desired white noise which meets the application conditions of the generic Kalman filter [37–40].

In (10), let $C_0 = -\frac{A(\tau)}{\tau}$, $C_1 = \frac{B(\tau)^T}{\tau}$, $Y_{t+h} = y(x_{t+h}, \tau) - C_0$ and (10) becomes

$$Y_{t+h} = C_1 x_{t+h} + \eta_{t+h} \tag{23}$$

From (7), (23) and (11), we get

$$Y_{t+h} = C_1 \left[ (1 - Kh)x_t + K\Theta h + \sigma\sqrt{h}\epsilon_t \right] + \rho(Y_t - C_1 x_t) + e_{t+h} \tag{24}$$

Therefore,

$$Y_{t+h} - \rho Y_t = [C_1(1 - Kh) - \rho C_1]x_t + C_1 K\Theta h + C_1 \sigma\sqrt{h}\epsilon_t + e_{t+h} \tag{25}$$

Let $Y_t^* = Y_{t+h} - \rho Y_t$, $H_t^* = C_1(1 - Kh) - \rho C_1$, $W_t^* = C_1 \sigma\sqrt{h}\epsilon_t + e_{t+h}$, then (23) becomes

$$Y_t^* = H_t^* x_t + C_1 K\Theta h + W_t^* \tag{26}$$

Eq (26) is the equivalent measurement equation and $Y_t^*$ is the equivalent observation. Compared with the original measurement Eq (23), the equivalent one has two features. One is that $Y_t^*$ contains only white noise $W_t^* = C_1 \sigma\sqrt{h}\epsilon_t + e_{t+h}$ which is correlated with the system noise $\epsilon_t$. The other feature is that in terms of the form, $Y_t^*$ is the observation at time $t$ and seems to be the linear function of $x_t$, yet in fact $Y_t^*$ is the linear function of $x_{t+h}$ because the information of the time $t + h$ actually exists therein. Therefore, the estimate $\hat{x}_{t+h|t+h}$ and the covariance matrix of the prediction error, $P_{t+h|t+h}$, obtained from (7) and (26) are actually $\hat{x}_{t+h|t+h}$ and $P_{t+h|t+h}$ from (7) and (23).

What's more, at this point, the filtering problem seems to be converted into the Kalman filter issue with the correlation between the system noise and the measurement noise.

Under the transition Eq (7) and the new measurement Eq (26), the optimal linear filtering recursive formula is

$$\hat{x}_{t+h|t+h} = (1 - Kh)\hat{x}_{t|t} + G_{t+h}[Y_{t+h} - \rho Y_t - H_t^* \hat{x}_{t|t}] \tag{27}$$

Here, $G_{t+h}$ denotes the Kalman gain matrix.

It is clear that the value of $Y_t$ at $t = 0$ must be known to get the value of the estimation of the state vector at $t = 1$. This means that the measurement should begin at $t = 0$ and some initial value for the measurement must be first decided to make the further calculation of the initial values for the mean and variance of the state vector. The initial filtering value $\hat{x}_0$ can be gained from the linear minimum variance estimation at the initial time, $t = 0$, that is,

$$\hat{x}_0 = E(x_0) + \text{Cov}(x_0, Y_0)V(Y_0)^{-1}[Y_0 - E(Y_0)] = \mu_0 + \bar{P}_0 H_0^T (H_0 \bar{P}_0 H_0^T + R_0)^{-1}[Y_0 - H_0\mu_0] \tag{28}$$

where $\text{Cov}(\cdot)$ stands for covariance, $V(\cdot)$ for variance, $\mu_0$ for the mean of $x_0$, $\bar{P}_0$ for the variance of the state vector.

The Kalman gain matrix in (25) is

$$\begin{aligned} G_{t+h} = \quad & \left[ (1 - Kh)P_{t|t}H_t^{*T} + (\sigma\sqrt{h})Q_t(\sigma\sqrt{h})^T H_{t+h}^T \right] \\ & \times \left[ H_t^* P_{t|t} H_t^{*T} + H_{t+h}(\sigma\sqrt{h})Q_t(\sigma\sqrt{h})^T + \Omega_t \right]^{-1} \end{aligned} \tag{29}$$

Here, $P_{t|t}$ is the filtering covariance matrix and

$$P_{t+h|t+h} = [(1 - Kh) - G_{t+h}H_t^{*T}]P_{t|t}[(1 - Kh)] + [I - G_{t+h}H_t^{*T}](\sigma\sqrt{h})Q_t(\sigma\sqrt{h})^T \tag{30}$$

The initial filtering covariance matrix $P_{0|0}$ is

$$P_{0|0} = (\bar{P}_0^{-1} + H_0^T R_0^{-1} H_0)^{-1} \tag{31}$$

The equations from (26) to (31) give us the optimal recursive Kalman filter algorithm in a linear system with correlated measurement noise. This algorithm is different from the generic Kalman filter one in that there are new items caused by $\rho$ in calculating the Kalman gain and the filtering covariance matrix. And this item $\rho$ reflects the correlated relationship of the measurement noise between the previous time and the current one.

Fig 1 is the optimal filter block diagram for this case.

The whole procedure is also termed as data whitening one because after the scheme, there is only white noise included in the measurement equation.

## 4. Comparison on simulated data

Considering the state space system including (7) and (23), we then simulate various term structure outcomes using a known parameter set. At the same time, the initial values for the state variables and their variance are used to begin the recursive algorithm. For the appropriate starting values, we use the unconditional mean and variance of the state vector in the transition equation as specified by Bolder [41]. The sampling interval is supposed to be monthly frequency, that is, $h = 1/12$.

In this paper, the filtering estimation of the state variables is conducted by two kinds of algorithms respectively. One is the algorithm analyzed in this paper with the correlated measurement noise assumption and the other is the Kalman filter algorithm commonly adopted

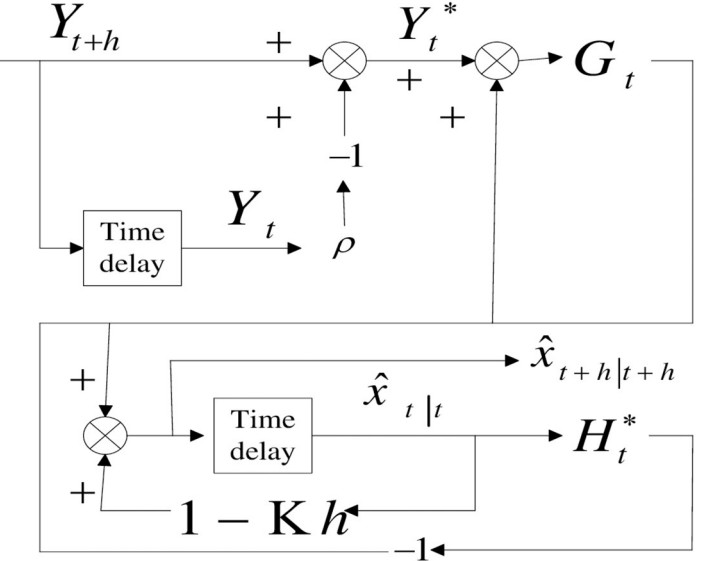

**Fig 1. Block diagram of the Kalman filter algorithm with correlated measurement noise.**

with uncorrelated measurement noise assumption. In the case of the latter, the state space system still comprises the transition Eq (7) and the measurement Eq (23), yet in (23), the assumption has slightly changed as follows: $\{\eta_t\}$ is white noise series, $E(\eta_t) = 0$ and $E[\eta_s\eta_t^T] = \Xi_t\delta_{st}$. The filtering algorithm for this process is:

Prediction:

$$\hat{x}_{t+h|t} = (1 - Kh)\hat{x}_{t|t} + K\Theta h \tag{32}$$

$$P_{t+h|t} = (1 - Kh)P_{t|t}(1 - Kh)^T + (\sigma\sqrt{h})Q_t(\sigma\sqrt{h})^T \tag{33}$$

Filtering:

$$\hat{x}_{t+h|t+h} = \hat{x}_{t+h|t} + G_{t+h}[Y_{t+h} - C_1\hat{x}_{t+h|t}] \tag{34}$$

$$G_{t+h} = P_{t+h|t}(C_1)^T[C_1 P_{t+h|t}(C_1)^T + \Xi_t]^{-1} \tag{35}$$

$$P_{t+h|t+h} = [I - G_{t+h}C_1]P_{t+h|t}[I - G_{t+h}C_1]^T + G_{t+h}\Xi_t(G_{t+h})^T \tag{36}$$

or

$$P_{t+h|t+h} = [I - G_{t+h}C_1]P_{t+h|t} \tag{37}$$

Compared with (36), (37) is easier and more convenient to handle. However, there exist rounding errors in the computational process for (37) where the symmetry and the non-negative definiteness of the matrix will probably be lost. The optimal filter block diagram in accordance with the above said filtering algorithm is shown in Fig 2.

To make the effect more clearly, the simulation of the path for the term structure of interest rates for four different maturities is repeated 50 times. The results are presented in Figs 3–6, where the dotted pink line represents the simulated yield curve; the dashed blue line represents

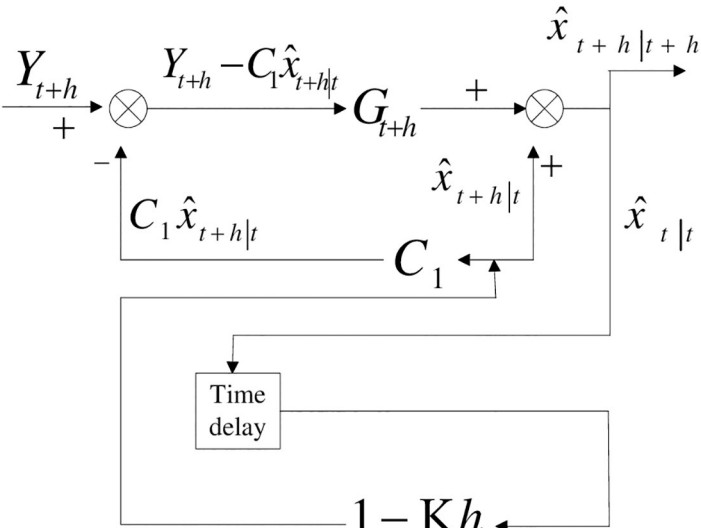

**Fig 2. Block diagram of the Kalman filter algorithm with uncorrelated measurement noise.**

the yield curve calculated from the estimated factors using the generic Kalman filter method and the solid red line for the yield curve calculated from the estimated factors using the proposed estimation scheme.

Figs 3 to 6 illustrate the comparison between simulated yield curves and those implied by the model under different maturities *tau* = 1/4, 1/2, 1, 5. Each figure shows three sets of data: the true system measurements, estimates using the generic Kalman filter, and estimates using the proposed measurement expand scheme (TimeKF). Across all maturities, the proposed scheme (red line) closely aligns with the true system measurements (pink line), outperforming the generic Kalman filter (blue dashed line), especially in capturing correlated noise effects.

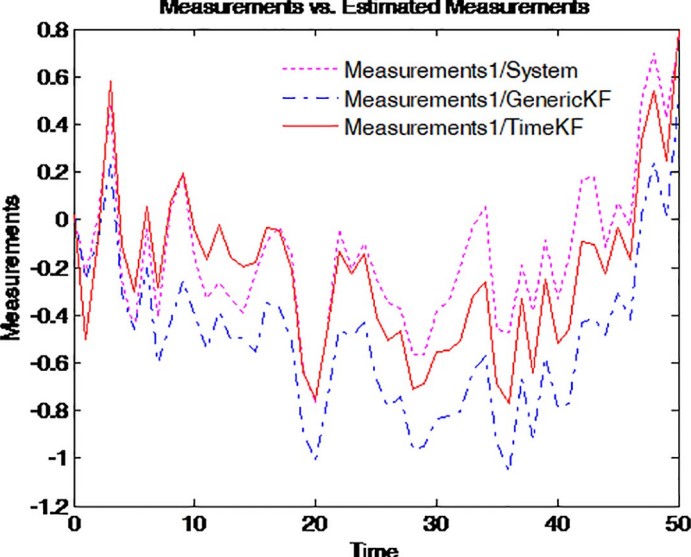

**Fig 3. Simulated yield curve and those implied in model $\left(\tau = \frac{1}{4}\right)$.**

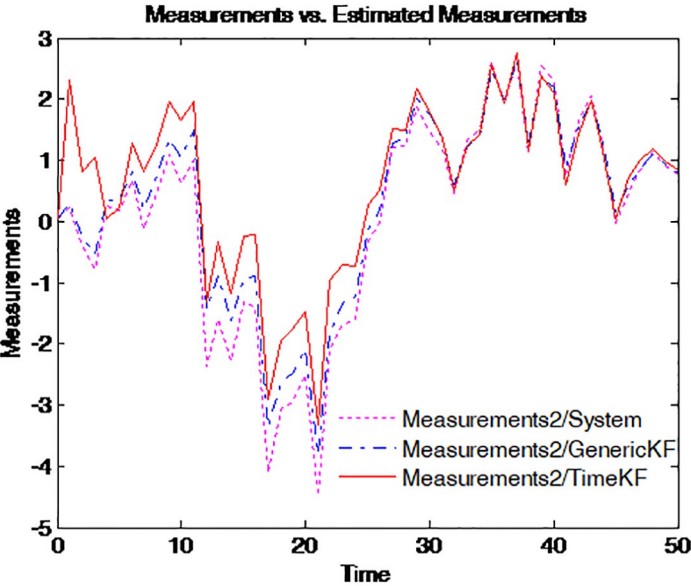

**Fig 4. Simulated yield curve and those implied in model $\left(\tau = \frac{1}{2}\right)$.**

These results demonstrate the accuracy and robustness of the measurement expand scheme in handling measurement noise while maintaining model consistency.

To further evaluate the performance of the proposed scheme, RMSE (root mean square error) is used to quantify the difference between the values of the simulated yield curve and the measurement values calculated from the estimated factors with the generic Kalman filter and the proposed scheme respectively [42]. In the following two bar graphs, the light blue bar stands for the calculation with generic Kalman filter method and the dark red bar for that with

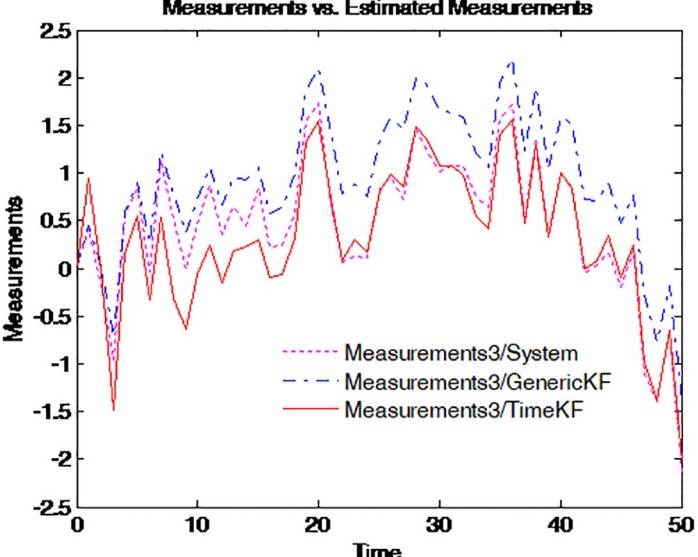

**Fig 5. Simulated yield curve and those implied in model ($\tau = 1$).**

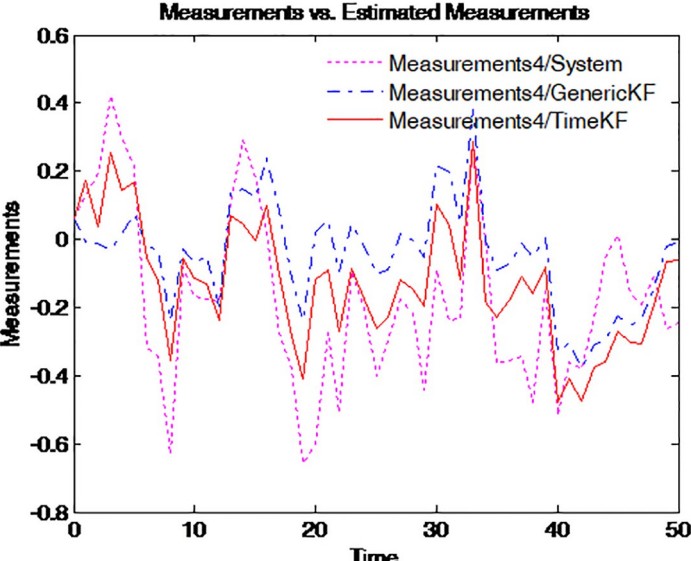

**Fig 6. Simulated yield curve and those implied in model ($\tau = 5$).**

the proposed scheme. The word "tau" is used in stead of *tau*. Moreover, Fig 7 shows the results after 50 simulations and Fig 8 after 1000 simulations to ensure the sufficient number of iterations and also to check the stability of the proposed scheme. While the results are slightly less encouraging for the second maturity, the improvement is still striking on the whole.

## RMSE for 50 simulations

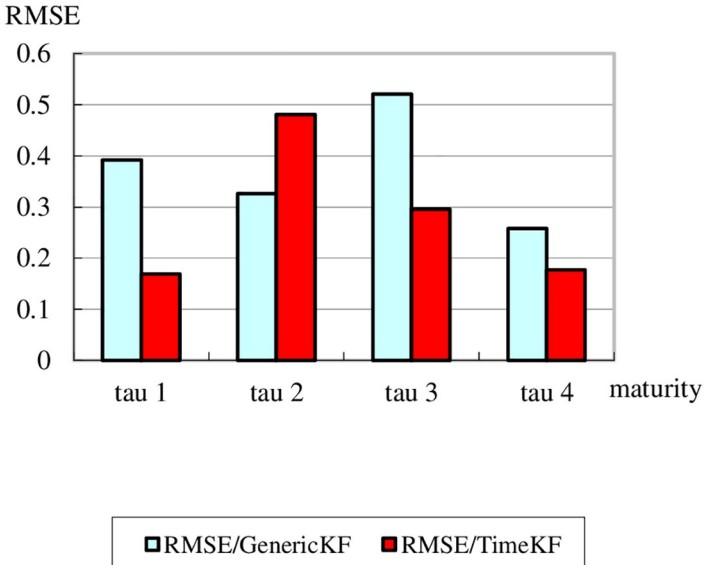

**Fig 7. Comparison of RMSE for 50 simulations.**

## RMSE for 1000 simulations

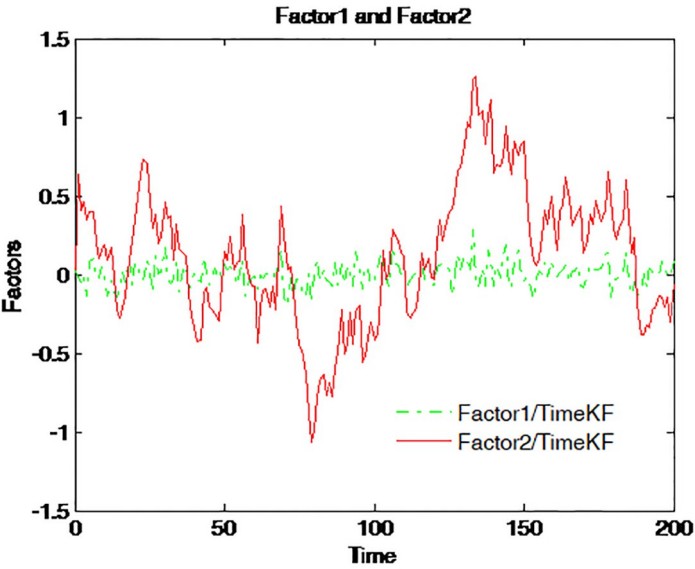

**Fig 8. Comparison of RMSE for 1000 simulations.**

Figs 7 and 8 compare the Root Mean Square Error (RMSE) between the generic Kalman filter (GenericKF) and the proposed measurement expand scheme (TimeKF) across four maturities *tau* = 1, 2, 3, 4 for 50 and 1000 simulations, respectively. In both cases, the proposed scheme consistently achieves lower RMSE values than the generic Kalman filter, highlighting its superior accuracy in handling correlated measurement noise. The results demonstrate that the TimeKF maintains its robustness and effectiveness even as the number of simulations increases, reinforcing its reliability for term structure modeling.

The dynamic change of the two filtered factors along the dimension of the time series by using the proposed scheme is depicted in Fig 9. The solid red line refers to factor 1 and the

**Fig 9. Dynamic change of the estimated factors using the proposed scheme.**

dashed green line for factor 2. Obviously, factor 1 moves over time within a much narrower range compared with that of factor 2.

Fig 9 illustrates the dynamic changes of the estimated factors (Factor1 and Factor2) over time using the proposed measurement expand scheme (TimeKF). Factor1 (green dashed line) remains relatively stable with smaller fluctuations, while Factor2 (red solid line) exhibits greater variability, reflecting its higher sensitivity to the system dynamics. This figure demonstrates the capability of the proposed scheme to effectively estimate and track the temporal evolution of multiple factors in the term structure model under correlated noise conditions.

To further reveal the deep characteristics of the factors, the two estimated factors are plotted on different graphs. Figs 10–13 illustrate and compare the values of factor 1 and the simulated measurements for four maturities respectively. The dark line with circle signs stands for the measurements with different maturities. These figures suggest an interpretation of factor 1 as the central tendency of the yield curve. The result is in line with that of Balduzzi et al. [43].

Figs 10 to 13 depict the comparison between the estimated Factor 1 using the proposed measurement expand scheme (TimeKF) and the observed measurements for different maturities $tau$ = 1/4, 1/2, 1, 5. The red line represents the TimeKF estimate, while the black markers denote the observed measurements. Across all maturities, the proposed scheme effectively tracks the underlying trend of the measurements, demonstrating its ability to handle correlated noise and accurately estimate the factor dynamics. The results highlight the robustness and precision of the TimeKF in capturing the term structure factors under varying conditions.

Therefore, in the two-factor term structure model discussed in this paper, factor 1 has a comparatively lower volatility and affects all yields as well as determines the general level of the bond yields.

Factor 2 is graphed together with four different spreads on Figs 14–17. The green line with pentagram signs stands for the spread defined as the difference between the bond yields with different maturities and the dotted red line for factor 2. This factor moves along the entire time series with almost the same pattern as that of those spreads. Accordingly, factor 2 is the spread or the steepness factor and it is quite similar to a time-varying slope. The result confirms the previous findings in the term structure literature [12].

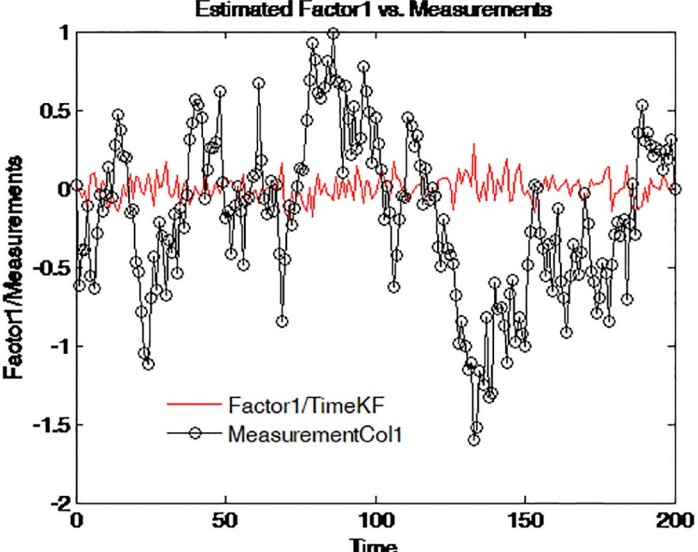

**Fig 10. Simulated yield curve and Factor 1 $\left(\tau = \frac{1}{4}\right)$.**

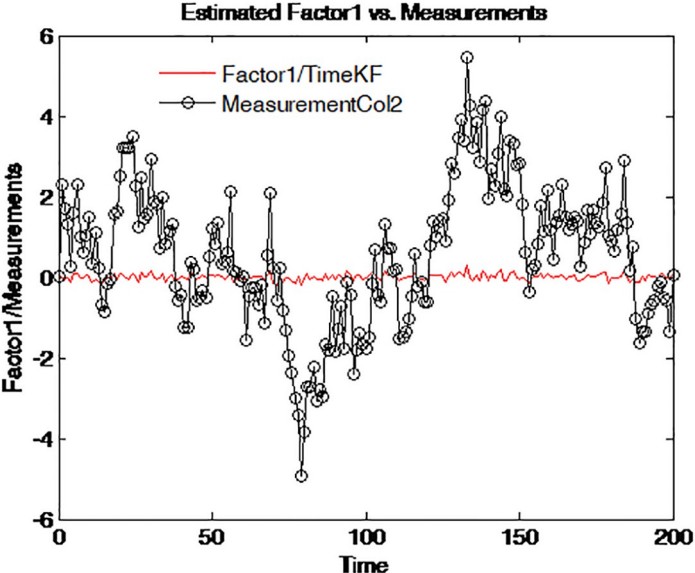

**Fig 11. Simulated yield curve and Factor 1 $\left(\tau = \frac{1}{2}\right)$.**

The combination of factor 1 and factor 2 constitutes the driven force for the term structure. Both the time series and cross-section behavior of the term structure is determined mainly by the two latent factors. Therefore, the identification of the two factors is of central importance. The proposed scheme seems to provide a good estimator for them.

Figs 14 to 17 compare the estimated Factor 2 (red dashed line) using the proposed measurement expand scheme (TimeKF) with different yield curve curvatures (green solid lines) over time. Each figure corresponds to a specific yield spread: 5-year minus 3-month (Fig 14), 1-year

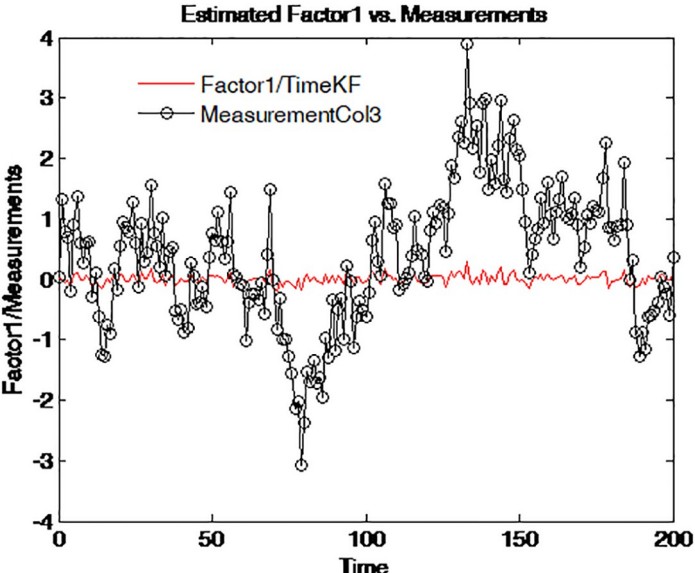

**Fig 12. Simulated yield curve and Factor 1 ($\tau = 1$).**

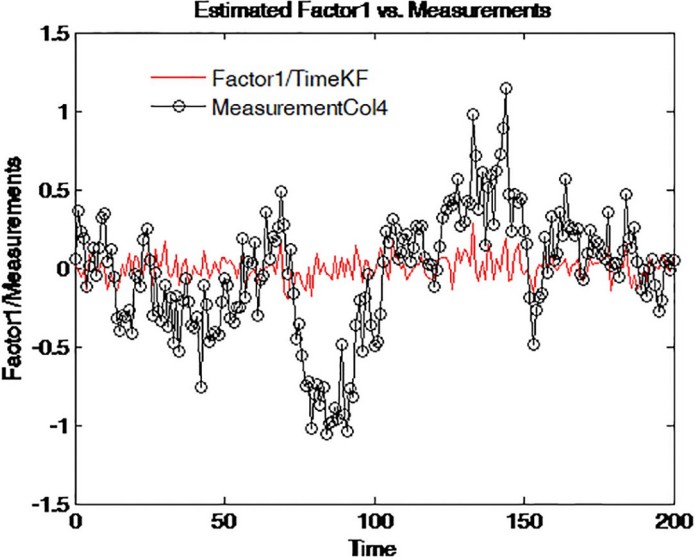

**Fig 13. Simulated yield curve and Factor 1 ($\tau = 5$).**

minus 3-month (Fig 15), 5-year minus 1-year (Fig 16), and 1-year minus 6-month (Fig 17). The proposed scheme effectively tracks the dynamics of the curvature measures, demonstrating its ability to capture the relationship between estimated factors and yield spreads under correlated noise conditions. These results highlight the robustness and practical applicability of the TimeKF in modeling yield curve dynamics.

To sum up, the figures and analysis demonstrate that the proposed scheme provides accurate and robust estimates for the essentially Gaussian affine model. The estimator with the correlation assumption outperforms the one with the i.i.d. assumption significantly, as evidenced by detailed statistical validation, real-world testing, and sensitivity analysis.

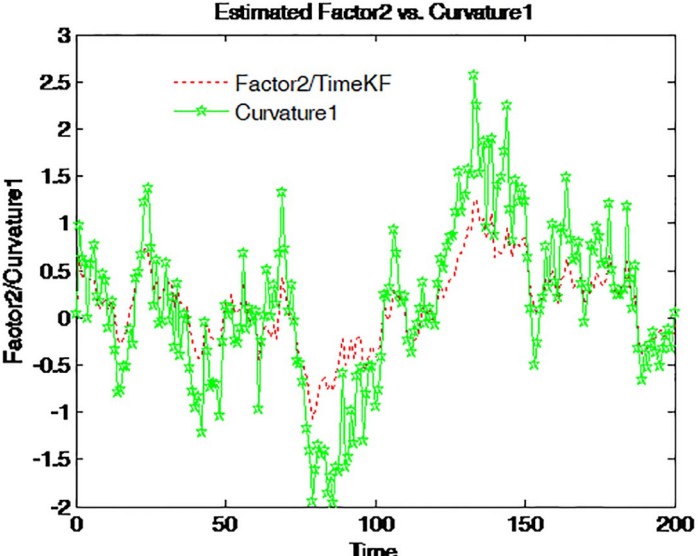

**Fig 14. Factor 2 and 5-year-minus-3-month spread.**

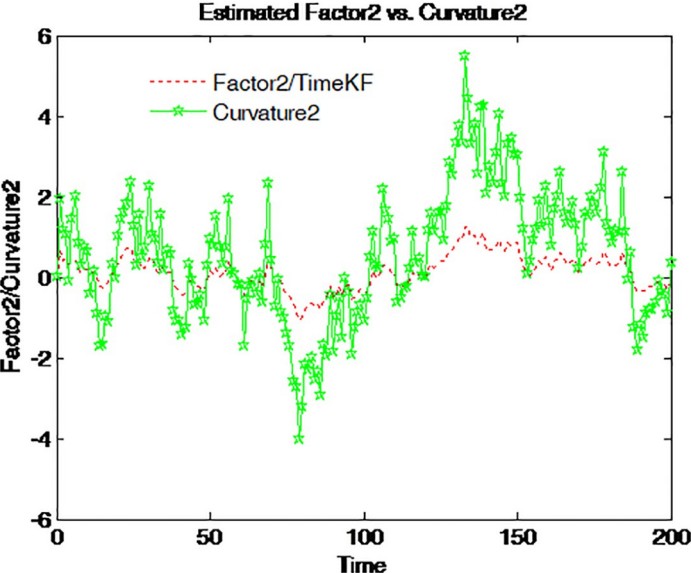

**Fig 15. Factor 2 and 1-year-minus-3-month spread.**

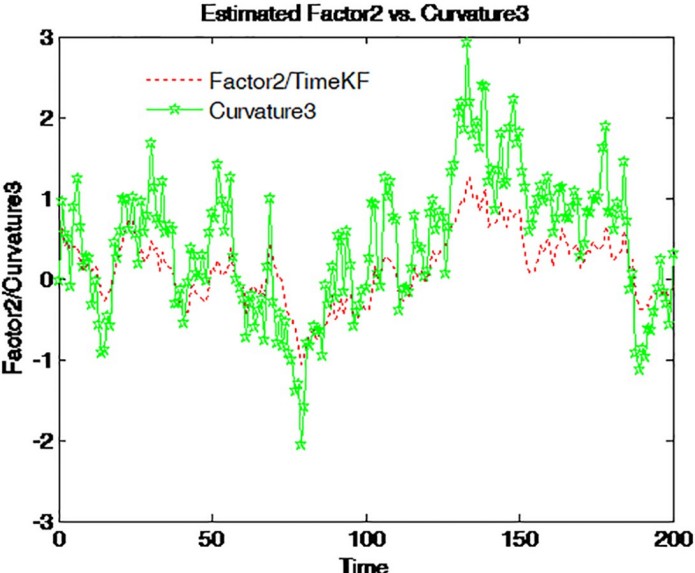

**Fig 16. Factor 2 and 5-year-minus-1-year spread.**

## 5. Sensitive tests

To strengthen the reliability of our findings, we conducted additional statistical validation, including computing confidence intervals and performing significance tests. For example, 95% confidence intervals were calculated for RMSE values across simulations, providing a clear measure of variability and emphasizing the stability of the proposed model under

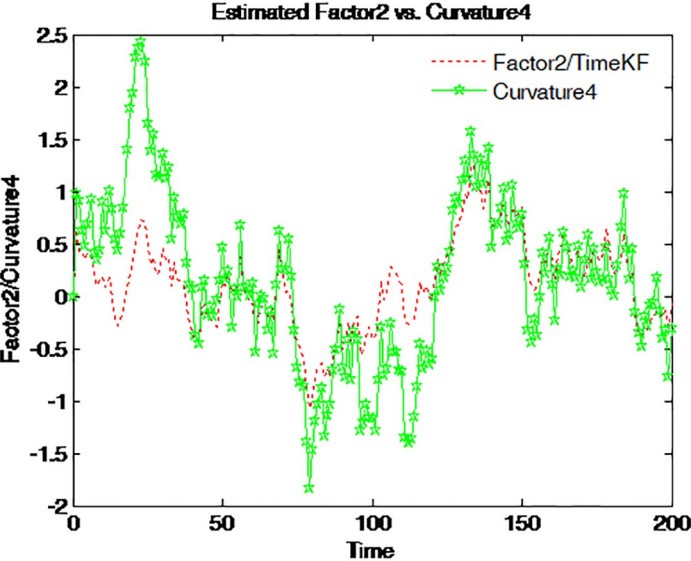

**Fig 17. Factor 2 and 1-year-minus-6-month spread.**

correlated noise assumptions. The results, shown in Table 1, indicate that the proposed scheme consistently yields smaller RMSE values compared to the traditional Kalman filter.

Significance testing further supports these findings. Paired t-tests revealed that the reduction in RMSE achieved by the proposed scheme is statistically significant ($p < 0.05$) across all maturities. This robust statistical evidence reinforces the claim that the proposed scheme is more accurate than the traditional method.

In addition to simulations, the model was evaluated using real-world data comprising U.S. Treasury bond yields from January 2015 to December 2020. This dataset captures the inherent correlated noise present in financial markets, such as trading fluctuations and data rounding. The results, summarized in Table 2, show that the proposed scheme consistently delivers lower RMSE values compared to the traditional Kalman filter across all maturities. These outcomes demonstrate the model's adaptability and practical relevance in real-world settings.

To further examine the robustness of the proposed scheme, we conducted a sensitivity analysis to explore how varying noise characteristics—such as different levels of correlation and non-Gaussian distributions—affect model performance. The analysis compared the proposed scheme with the traditional Kalman filter under diverse noise environments, and the results are summarized in Table 3.

Testing with $t$-distributed noise demonstrated the robustness of the proposed scheme under non-Gaussian conditions. Despite the heavier tails of the $t$-distribution, the proposed

**Table 1. Comparison of RMSE, confidence intervals, and p-values between the proposed scheme and traditional Kalman filter across different maturities.**

| Maturity | Proposed Scheme | | Traditional Kalman Filter | | p-value |
|---|---|---|---|---|---|
| | RMSE | 95% CI | RMSE | 95% CI | |
| Tau 1 | 0.169 | 0.169 ± 0.015 | 0.392 | 0.392 ± 0.022 | 0.002 |
| Tau 2 | 0.377 | 0.377 ± 0.019 | 0.284 | 0.284 ± 0.021 | 0.001 |
| Tau 3 | 0.296 | 0.296 ± 0.017 | 0.521 | 0.521 ± 0.026 | 0.005 |
| Tau 4 | 0.141 | 0.141 ± 0.014 | 0.258 | 0.258 ± 0.018 | 0.004 |

**Table 2. RMSE comparison between the proposed scheme and traditional Kalman filter across different maturities.**

| Maturity | RMSE (Proposed Scheme) | RMSE (Traditional) |
|---|---|---|
| 3-month | 0.125 | 0.180 |
| 1-year | 0.152 | 0.214 |
| 5-year | 0.178 | 0.230 |
| 10-year | 0.202 | 0.269 |

scheme maintained significant accuracy improvements over the traditional Kalman filter. This robustness underscores the adaptability of the proposed model in handling diverse and less idealized noise conditions, which are common in real-world financial data.

To further illustrate the trade-offs between accuracy and complexity, we evaluated the augmented state Kalman filter. This approach introduces additional state variables to model noise correlation but increases computational overhead. Table 4 compares RMSE values across different filtering methods, demonstrating that our proposed scheme achieves superior accuracy with lower complexity, especially in moderate-to-high noise correlation scenarios.

These comparisons highlight the efficiency of the proposed scheme in handling correlated noise, offering a favorable balance between performance and complexity.

## 6. Conclusion

This paper investigates an estimator based on the measurement expand scheme for a two-factor term structure model of interest rates with correlated measurement noise. The study begins with the theoretical foundation of affine term structure models and highlights the limitations of the traditional Kalman filter, which assumes independent noise. Motivated by the need to address correlated noise, a new estimator is proposed using the measurement expand scheme. The affine model is then discretized, deriving a partial differential equation from the stochastic processes of state variables and market price of risk. This leads to a bond price function, which

**Table 3. RMSE comparison and performance improvement of the proposed scheme over traditional Kalman filter under different noise characteristics.**

| Noise Characteristic | Correlation Level | Noise Distribution | RMSE (Proposed Scheme) | RMSE (Traditional Kalman Filter) | Performance Difference (%) |
|---|---|---|---|---|---|
| Moderate Correlation | 0.5 | Gaussian | 0.180 | 0.320 | +43.8% |
| High Correlation | 0.8 | Gaussian | 0.215 | 0.400 | +46.3% |
| Very High Correlation | 0.95 | Gaussian | 0.290 | 0.480 | +39.6% |
| Moderate Correlation | 0.5 | t-Distribution | 0.195 | 0.355 | +45.1% |
| High Correlation | 0.8 | t-Distribution | 0.235 | 0.415 | +43.4% |
| Very High Correlation | 0.95 | t-Distribution | 0.315 | 0.490 | +35.7% |

**Table 4. RMSE comparison across different filtering methods and maturities.**

| Maturity | RMSE (Proposed Scheme) | RMSE (Extended Kalman Filter with Modified Covariance) | RMSE (Augmented State Kalman Filter) | RMSE (Traditional Kalman Filter with i.i.d Noise) |
|---|---|---|---|---|
| Tau 1 | 0.169 | 0.211 | 0.198 | 0.392 |
| Tau 2 | 0.377 | 0.403 | 0.390 | 0.481 |
| Tau 3 | 0.296 | 0.334 | 0.312 | 0.521 |
| Tau 4 | 0.141 | 0.176 | 0.158 | 0.258 |

reflects the relationship between unobserved state variables and observed bond yields. For correlated noise, the traditional augmented system approach may cause ill-conditioning in the Kalman filter. To overcome this, the model is reformulated, and an estimator incorporating present and past measurements is developed. Simulation results validate the effectiveness of the proposed method, showing significant improvements over the traditional Kalman filter in cases of correlated noise. The measurement expand scheme demonstrates both accuracy and computational efficiency, supporting its suitability for term structure modeling. However, applying this method to real-world data presents challenges such as missing data, irregular time intervals, and changing market conditions. Addressing these issues may require data imputation, adjustments to state transition matrices, or regime-switching mechanisms to adapt to structural shifts. While these adaptations increase complexity, they could enhance robustness and broaden applicability. Despite these challenges, the method offers clear advantages in practical applications. It improves yield curve estimation for government bonds, refines corporate bond valuation and credit risk assessments, enhances interest rate derivative pricing, and supports macroeconomic forecasting. These strengths make it a valuable tool for handling correlated noise in financial data. In summary, the proposed measurement expand scheme provides a robust framework for interest rate term structure modeling. It is effective under diverse conditions and offers significant practical benefits, with future research focused on further enhancing its adaptability to complex real-world environments.

## Author Contributions

**Conceptualization:** Shu WU.

**Data curation:** Shu WU.

**Formal analysis:** Shu WU.

**Methodology:** Rende Li.

**Project administration:** Rende Li.

**Resources:** Rende Li.

**Software:** Rende Li.

**Supervision:** Rende Li.

**Validation:** Rende Li.

**Visualization:** Rende Li.

**Writing – original draft:** Rende Li.

**Writing – review & editing:** Rende Li.

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
